# Phenotypic characterization of single CD4+ T cells harboring genetically intact and inducible HIV genomes

Caroline Dufour[1], Corentin Richard[1], Marion Pardons[1], Marta Massanella [1], Antoine Ackaoui[1], Ben Murrell[2], Bertrand Routy [1], Réjean Thomas[3], Jean-Pierre Routy [4], Rémi Fromentin [1] & Nicolas Chomont [1] ✉

The phenotype of the rare HIV-infected cells persisting during antiretroviral therapies (ART) remains elusive. We developed a single-cell approach that combines the phenotypic analysis of HIV-infected cells with near full-length sequencing of their associated proviruses to characterize the viral reservoir in 6 male individuals on suppressive ART. We show that individual cells carrying clonally expanded identical proviruses display very diverse phenotypes, indicating that cellular proliferation contributes to the phenotypic diversification of the HIV reservoir. Unlike most viral genomes persisting on ART, inducible and translation-competent proviruses rarely present large deletions but are enriched in defects in the Ψ locus. Interestingly, the few cells harboring genetically intact and inducible viral genomes express higher levels of the integrin VLA-4 compared to uninfected cells or cells with defective proviruses. Viral outgrowth assay confirmed that memory CD4+ T cells expressing high levels of VLA-4 are highly enriched in replication-competent HIV (27-fold enrichment). We conclude that although clonal expansions diversify the phenotype of HIV reservoir cells, CD4+ T cells harboring replication-competent HIV retain VLA-4 expression.

Latent HIV genomes permanently integrated in a small pool of CD4+ T cells persist during antiretroviral therapy (ART) and represent the major obstacle to a cure[1–4]. ART abrogates HIV replication but has no impact on latently infected cells and hence, does not eradicate the virus. HIV latency is characterized by multiple and non-mutually exclusive features, including proviral integration into less transcriptionally active regions of the chromatin[5–8], lack of essential transcription and elongation cellular factors[9,10] and moderate activation status of the infected cell[11], which all contribute to HIV silencing. Nevertheless, viral rebound occurs shortly after ART interruption in most people living with HIV (PLWH)[12], indicating that productive infection can reignite from latent viral reservoirs.

The half-life of the replication-competent viral reservoir in PLWH on ART has been estimated to 44 months[13–16], and this longevity is attributed to the survival, self-renewal and proliferative capacities of infected memory CD4+ T cells[17–22]. Accumulating evidence indicates that cell proliferation plays a major role in the persistence of HIV reservoirs. Using sequencing of integration sites[21–24], near full-length (NFL) proviruses[25–30], T cell receptors (TCR)[31–33] or combinations of these approaches[33–37], numerous studies have highlighted the major contribution of clonal expansion to HIV persistence. Since clonal expansion of CD4+ T cells harboring viral genomes does not necessarily lead to viral production nor to the clearance of the infected cell[38], sequential waves of antigen-induced expansions and contractions

[1]Centre de Recherche du CHUM and Department of Microbiology, Infectiology and Immunology, Université de Montréal, Montreal H2X 0A9 Quebec, Canada. [2]Department of Microbiology, Tumor and Cell Biology, Karolinska Institutet, Stockholm 171 77, Sweden. [3]Clinique médicale l'Actuel, Montreal H2L 4P9 Quebec, Canada. [4]Division of Hematology & Chronic Viral Illness Service, McGill University Health Centre, Montreal H4A 3J1 Quebec, Canada. ✉e-mail: nicolas.chomont@umontreal.ca

result in dynamic changes within the reservoir during ART[32]. Although it is clear that these sequential episodes of expansions influence the genetic diversity of the proviral populations, their impact on the phenotypic diversity of the cells carrying persistent HIV remains unknown[39]. As such, whether individual cells belonging to a single proviral clone display similar phenotypes is unclear.

Several cellular markers that enrich in HIV reservoirs have been identified (reviewed in[40]). Within the CD4+ T cell compartment, HIV is enriched in cells displaying a central ($T_{CM}$) and effector memory ($T_{EM}$) phenotype[18–20,25,41–44], in cells expressing the immune checkpoint molecules PD-1 (programmed death-1), TIGIT (T-cell immunoglobulin and ITIM domain) and LAG-3 (lymphocyte activation gene 3)[18,45–48], as well as the activation markers HLA-DR and ICOS[49,50]. Other cell-surface markers enriching in persistent HIV genomes include the adhesion molecule CD2[51], the TNF receptor CD30[52], the FC gamma receptor CD32[53,54], the homing receptor VLA-4[50] and the B cell marker CD20[55]. Additional discrete CD4+ T cell subsets enriched in HIV-infected cells have been described, including stem cell memory T cells ($T_{SCM}$)[19,20], regulatory T cells ($T_{reg}$)[56–58], T follicular helper cells ($T_{FH}$)[46,47], and functionally polarized Th1[59], Th2[30] and Th17 cells[60]. However, most of these studies used HIV DNA measures as surrogate of the HIV reservoir, an approach that is known to largely overestimate the size of the replication-competent reservoir, since most HIV genomes persisting during ART are genetically defective[25,59,61–63]. While a significant proportion of these proviruses retain the capacity to transcribe viral RNAs and to produce viral proteins[8,27,50,64,65], only a small fraction can produce infectious virions and represent a potential source of viral rebound. If the genetic intactness of these proviruses is a prerequisite to their ability at causing rebound, their inducibility is also critical but has rarely been evaluated[15,25,26,59,62,66], (reviewed in[67]). As such, genetically intact genomes persisting in a deeply latent state and which are difficult to reactivate may be less clinically relevant than those from which viral production can be induced[8,68,69].

Here, we used a combination of approaches to obtain the phenotype of CD4+ T cells harboring genetically intact and inducible HIV proviruses in individuals on prolonged ART.

## Results

### Combined phenotypic and genotypic analysis of the inducible and translation-competent reservoir

To analyze the genetic integrity of inducible HIV genomes persisting in latently infected cells and assess their phenotypes, we developed a single-cell approach that combines indexed single-cell sorting of p24+ cells (HIV-Flow, which allows the isolation of single cells with retrospective identification of each single cell's high-dimensional immune phenotype[50]) with near full-length (NFL) proviral sequencing (Fig. 1a). To obtain NFL HIV sequences from single sorted cells, we modified the FLIPS assay originally described by Hiener et al.[25] by designing primers with degenerated nucleotides to capture a larger proportion of HIV genomes, by using a novel high-fidelity enzyme that maximizes long-range amplification with minimal error-rate, and by using the PacBio next-generation sequencing platform which provides excellent fidelity and avoids the need for sequence reconstruction. We first evaluated the efficacy and fidelity of our sequencing approach by sorting single ACH-2 cells (harboring a single copy of HIV provirus[70]) and amplified their viral genomes using our optimized protocol (Fig. S1a). NFL genomes were successfully amplified in 71% of the wells containing a single HIV-infected ACH-2 cell. PacBio sequencing yielded extremely low error rate: 1 in 34,025 nucleotides, with 97% of NFL sequences showing 100% identity (Fig. S1a). We used this approach to obtain the phenotype and proviral sequences of HIV-infected cells harboring inducible proviruses in blood samples from 6 PLWH on suppressive ART for a median of 9.5 years (Table 1). PMA/ionomycin was used to induce p24 production from latent proviruses. The expression levels of 8 cellular markers previously described to enrich in HIV-infected cells (PD-1,

TIGIT, HLA-DR, ICOS, α4, β1, CD45RA and CCR7[18,45,49,50]) and which expression were not modified by the stimulation in the presence of brefeldin A (BFA), were simultaneously recorded (Fig. 1b and S1b). We obtained a total of 308 NFL sequences from single-sorted p24+ cells (range: 41–59 sequences per participant). To analyze the proviral landscape of the non-induced and/or translation-incompetent reservoir in these same individuals, we also obtained 326 sequences from serially diluted p24−negative cells (range: 24–76 sequences per participant). We constructed phylogenetic trees containing genomic sequences derived from p24+ and p24− cells (Fig. 2a–f). In all 6 participants, proviral sequences retrieved from p24+ cells and p24− cells were intermingled, indicating that inducible and translation-competent proviruses do not cluster separately from viral genomes that cannot produce p24 upon reactivation.

### The inducible and translation-competent reservoir is highly clonal

Since clonal expansion is a major mechanism by which latently infected CD4+ T cells persist during ART[23,31–34,71], we estimated the contribution of cell proliferation to the maintenance of the inducible and non-induced reservoirs by determining the proportions of duplicated proviral sequences in both compartments. Clonal expansions of HIV-infected cells were detected in both the p24+ and p24− fractions in all participants (Fig. S2a). Duplicated HIV genomes were more frequently found in p24+ cells compared to p24− cells (means 78.9% vs 50.0%, respectively, $p < 0.0001$) (Fig. 3a). In line with this observation, large clonal expansions of HIV genomes (≥5 copies per clone) were found at a higher proportion in p24+ cells when compared to their negative counterparts (40.7% vs 11.4%, respectively, Fisher's exact test $p = 0.0073$) (Fig. 3b). Interestingly, proviral sequences shared between the induced and non-induced reservoirs were frequently observed (16 clones in 5/6 participants, Fig. 3c), indicating that within an infected T cell clone, individual genomes can vary in their ability to get reactivated upon a single round of stimulation. Altogether, these results highlight the prominent role of T cell proliferation in the maintenance of the inducible and translation-competent HIV reservoir, suggesting that these cells can evade immune recognition during this process. In addition, the observation that genetically identical proviruses are often found in both p24+ and p24− cells indicates that inducibility does not solely depends on the viral sequence and suggests that cellular features may influence the inducibility of a latent provirus.

### Clonally-expanded inducible proviruses are found in phenotypically different cells

Having demonstrated that the inducible and translation-competent reservoir is highly clonal, we sought to identify a phenotypic signature of these expanded proviral populations. For each individual p24+ cells, we linked the level of expression of the 8 cellular markers recorded during cell sorting (PD-1, TIGIT, HLA-DR, ICOS, α4, β1, CD45RA, and CCR7) to their cognate proviral sequence (obtained by NFL genome sequencing). For each cellular marker, we calculated the ratio between the fluorescence intensity measured on a single p24+ cell ($n = 239$) and the mean fluorescence intensity (MFI) of all CD4+ T cells from the same participant. In line with previous studies[18,45,50], p24+ cells tended to express higher levels of α4 and β1 (mean ratio of 2.17 and 1.26, respectively) and lower levels of CD45RA and CCR7 when compared to the global pool of CD4+ T cells (Fig. 4a and S3a). Unexpectedly, individual p24+ cells belonging to a given clonally expanded population expressed highly variable levels of most cellular markers analyzed (Fig. 4b), with the highest variations observed for α4, β1 and CCR7 (standard deviation of 1.18, 0.42 and 0.38, respectively, Fig. 4a and S3b). In line with previous studies[25,31,49], clonally expanded proviruses were less frequently found in $T_{CM}$ cells (65.4%), compared to $T_{EM}$ cells (81.4%; $p = 0.0041$) and in cells expressing TIGIT (86.5%, $p = 0.006$), HLA-DR (84.8%; $p = 0.0077$) and α4β1 (78.1%; $p = 0.025$) (Fig. 4c).

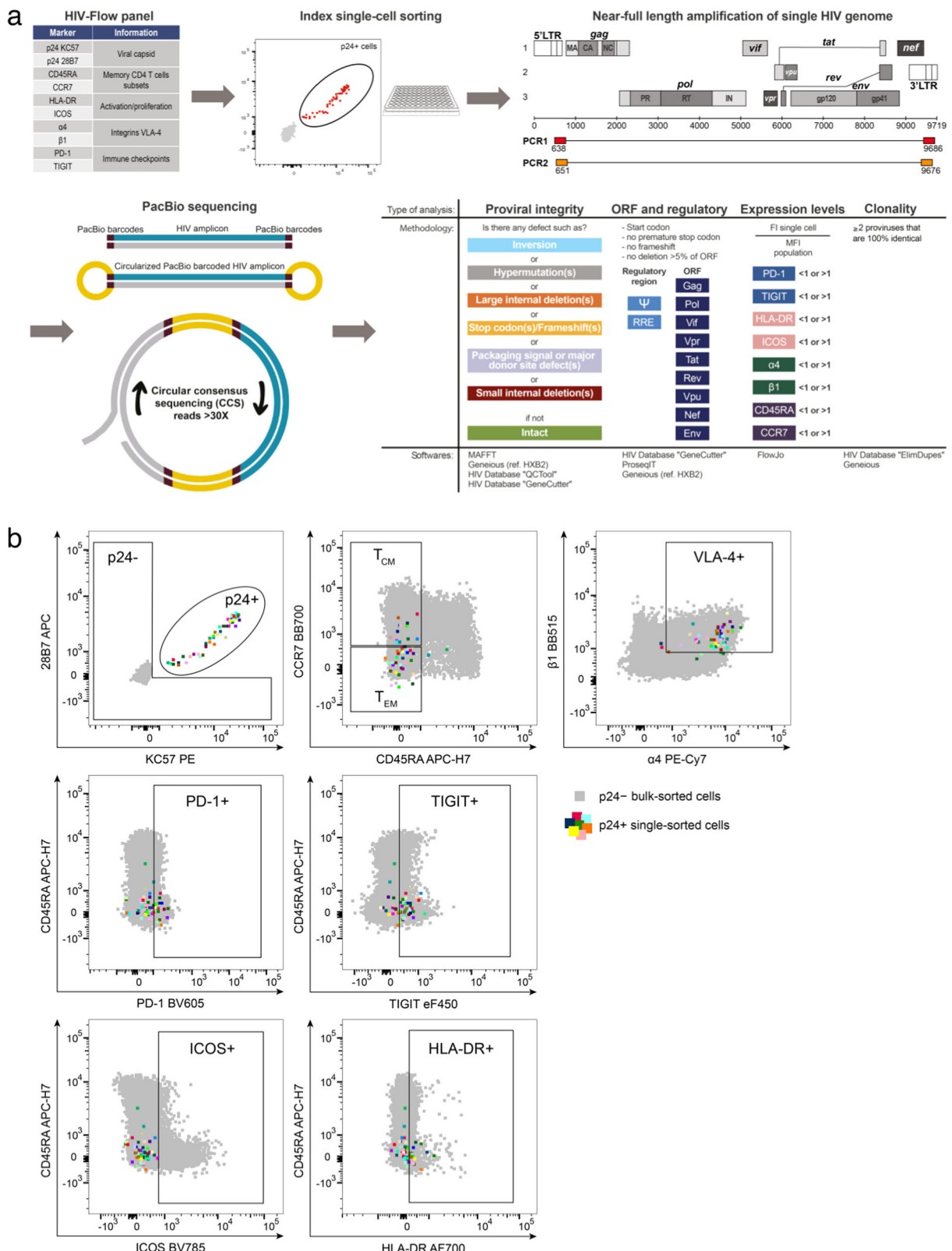

**Fig. 1 | Near full-length genome sequencing in single p24-expressing cells.**
**a** Isolated CD4+ T cells were stimulated with PMA/ionomycin for 24 h in the presence of BFA prior to extracellular and intracellular staining using antibodies specific to the cellular and viral proteins listed in the HIV-Flow panel table. Single p24+ cells were index-cell sorted in individual wells. NFL HIV amplification by nested PCR was performed on each single sorted p24+ cells, followed by PacBio

next-generation sequencing. Proviruses were analyzed individually for their genetic integrity. Levels of expression of cellular markers on each individual cell was retrieved and analyzed according to the genetic integrity and clonality of the provirus from this cell. **b** Representative dot plots of an HIV-Flow staining from participant ART5. Each colored dot represents a single infected cell (p24+) that was index-sorted. p24+ cells are overlaid onto p24− cells in grey.

**Table 1 | Cohort characteristics. Clinical data of the 10 ART-treated participants at the time of blood collection**

|  | ART1 | ART2 | ART3 | ART4 | ART5 | ART6 | ART7 | ART8 | ART9 | ART10 | Median |
|---|---|---|---|---|---|---|---|---|---|---|---|
| Characteristics of participants |  |  |  |  |  |  |  |  |  |  |  |
| Sex | M | M | M | M | M | M | M | M | M | M |  |
| Age, year-range | 50-59 | 60-69 | 60-69 | 40-49 | 30-39 | 50-59 | 50-59 | 30-39 | 60-69 | 30-39 | 53.5 |
| Time since diagnosis, years | 29 | 23 | 26 | 15 | 12 | 21 | 14 | 7 | 5 | 14 | 15 |
| Time on antiretroviral treatment, years | 14 | 17 | 21 | 14.8 | 4 | 20 | 6 | 6.5 | 4.3 | 9 | 12 |
| Time between diagnosis and ART initiation, years | 15 | 7 | 5 | 0.3 | 8 | 1 | 8 | 0.5 | 0.4 | 5 | 5 |
| HIV viral load, RNA copies per ml of plasma | <40 | <40 | <40 | <40 | <40 | <40 | <40 | <40 | <40 | <40 | <40 |
| CD4 cell count, cells per µL | 847 | 602 | 1076 | 625 | 471 | 911 | 836 | 461 | 624 | 882 | 731 |
| CD8 cell count, cells per µL | 613 | 880 | 558 | 511 | 322 | 873 | 1551 | 941 | 545 | 1177 | 743 |
| CD4/CD8 ratio | 1.4 | 0.7 | 1.9 | 1.2 | 1.5 | 1.0 | 0.5 | 0.5 | 1.1 | 0.8 | 1.1 |
| Integrated HIV DNA, copies per million CD4 T cells | 403 | 1404 | 442 | 438 | 146 | 1913 | 680 | 502 | 142 | 483 | 463 |
| HIV-Flow, p24+ cells per million CD4 T cells | 8 | 6 | 11 | 9 | 15 | 10 | 0 | 5 | 3 | 8 | 8 |
| Assays performed |  |  |  |  |  |  |  |  |  |  |  |
| Single-cell sorting assay | X | X | X | X | X | X |  |  |  |  |  |
| VLA-4 HIV DNA measures | X | X | X | X |  | X | X | X | X | X |  |
| VLA-4 QVOA |  | X | X |  |  | X | X | X | X |  |  |

*M* male, *ART* antiretroviral therapy, *QVOA* quantitative viral outgrowth assay.

Altogether, our results indicate that while clonal expansions are primarily detected in differentiated and activated CD4+ T cells, identical proviruses are found in p24+ cells displaying very diverse phenotypes, suggesting that clonal expansion contributes to the phenotypic diversification of HIV reservoir cells.

**Inducible and translation-competent proviruses display distinct defects compared to non-induced proviruses**

We then compared the integrity of HIV genomes from the inducible (Fig. 5a) and non-induced reservoirs (Fig. 5b). Proviruses retrieved from p24+ cells were significantly longer than those isolated from p24− cells (median lengths of 9027 bp and 4,804 bp, respectively, $p < 0.0001$), reflecting the frequent large deletions observed in non-induced proviruses (Fig. 5c). We then evaluated integrity by interrogating sequentially all proviral sequences for the presence of different genetic defects: Each provirus was analyzed for the presence of inversion, hypermutations, large internal deletions, deleterious stop codons in any of the coding regions apart from *nef*, defects in the packaging signal including deletion in one of the 4 stem loops or a point mutation in the major splice donor (MSD) site, and finally internal deletions larger than 5% of the ORF (Fig. 1a). Proviral sequences devoid of any of these defects were considered genetically intact[25,44]. Defects displayed by HIV genomes isolated from p24− cells were similar to what was previously reported[25,59,62] with 77.6% of the viral sequences with large internal deletions, 8.0% with hypermutation, 6.8% with defects in the Ψ locus, 2.8% with inversions and 2.8% with deleterious stop codons. In sharp contrast, proviruses retrieved from the inducible and translation-competent (p24+) reservoir did not contain inversion and hypermutation, and rarely presented large internal deletion (20.7%). However, 61.4% of the genomes retrieved from p24+ cells displayed mutations in the Ψ locus, a proportion that was significantly higher than in p24− cells (6.7%, Fisher's exact test $p < 0.0001$) (Fig. 5d). These defects corresponded to point mutations in the MSD site (8.5% in p24+), deletion of stem loop 2 (34.9% in p24+) or larger deletion of the Ψ locus (56.6% in p24+) (Fig. S2d). Since gag proteins are translated from unspliced viral mRNA[72], Ψ or MSD defects should not preclude their expression. Surprisingly, among the 94 unique proviral sequences retrieved from p24+ cells, 35 lacked the initial start codon in *gag* (position 790 in HXB2), an observation also made by Cole et al.[35]. In every of these sequences, we identified a second ATG located further, which likely substitutes for this deletion

(Fig. S4a). Of note, 3 of these "later *gag* start" sequences had premature stop codons 9 nucleotides upstream of the original *gag* stop codon, which did not impact p24 translation.

Genetically intact proviral sequences were rare and detected in p24+ cells from 3 of the 6 participants and in p24− cells from only 1 participant (Fig. S2b). Overall, the proportion of genetically intact proviruses was comparable between p24+ and p24− cells (3.9% and 2.1%, respectively) (Fig. 5d). Several of these intact proviruses were found in multiple copies, indicating that inducible and intact proviruses have the ability to persist through proliferation during ART (Fig. S2c), consistent with prior studies[38,73].

To further compare the genetic intactness of HIV proviruses in p24+ and p24− cells, we assessed the integrity of all 9 coding regions (*gag, pol, vif, vpr, tat, rev, vpu, nef, env*) and 2 regulatory regions (Ψ and RRE) independently (Fig. 1a). In all 6 participants, the vast majority of p24+ cells harbored intact viral genes coding for *vif, vpr, tat, rev, vpu* and *env*, and a complete and putatively functional RRE, whereas defects in all these regions were frequently observed in p24− cells (Fig. 5e). The *pol, nef* and to a lesser extent *gag* genes, also tended to be more frequently intact in p24+ cells. However, the majority of p24+ cells had genetic defects in the Ψ region (mean = 90.9%), a defect that was rarely observed in p24− cells (mean = 38.9%, $p = 0.031$).

We conclude that inducible and translation-competent proviruses are genetically distinct from their non-induced and/or translation-incompetent counterparts. Although viral genomes retrieved from p24+ cells are rarely intact, they show fewer defects, which are mostly located in the Ψ region.

**p24+ cells with intact proviral genomes express high levels of the integrin α4β1**

We then sought to identify the phenotypic signature of cells harboring genetically intact HIV genomes. The levels of expression of PD-1, ICOS, and CD45RA did not differ between cells harboring intact and potentially replication-competent proviruses and those carrying proviruses with large deletions, stop codons or Ψ defects (Fig. 6a). In contrast, cells with intact HIV proviruses expressed lower levels of the immune checkpoint molecule TIGIT compared to cells harboring genomes with Ψ defects ($p = 0.018$). CD4+ T cells with intact and inducible HIV genomes expressed higher levels of HLA-DR ($p = 0.033$) and lower levels of CCR7 ($p = 0.025$) than cells carrying genomes with stop codons. Strikingly, the integrin subunit α4 (CD49d) was expressed at higher

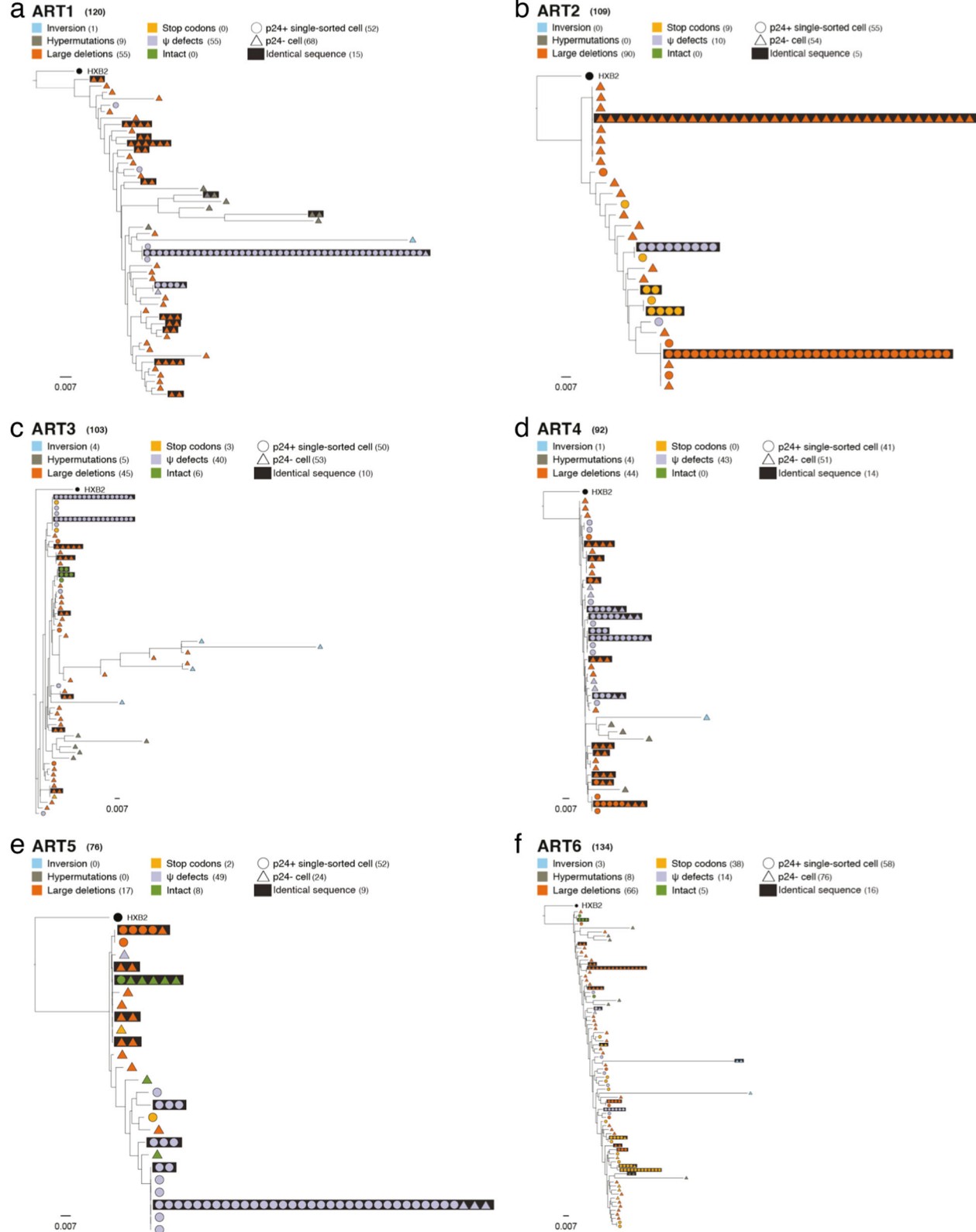

**Fig. 2 | Phylogenetic analysis of HIV genomes in p24+ and p24− populations from 6 ART-treated participants.** Maximum-likelihood phylogenetic trees of proviral genomes retrieved from p24+ cells (dot) and p24− cells (triangle) from the 6 participants: **a** ART1, **b** ART2, **c** ART3, **d** ART4, **e** ART5 and **f** ART6. Each genetic integrity/defect category is color-coded. Clonal expansions of identical proviruses are framed in black rectangles. The total number of proviruses sequenced for each participant are indicated at the top of each tree next to each category. The HXB2 sequence (black dot) was included as a reference in each tree.

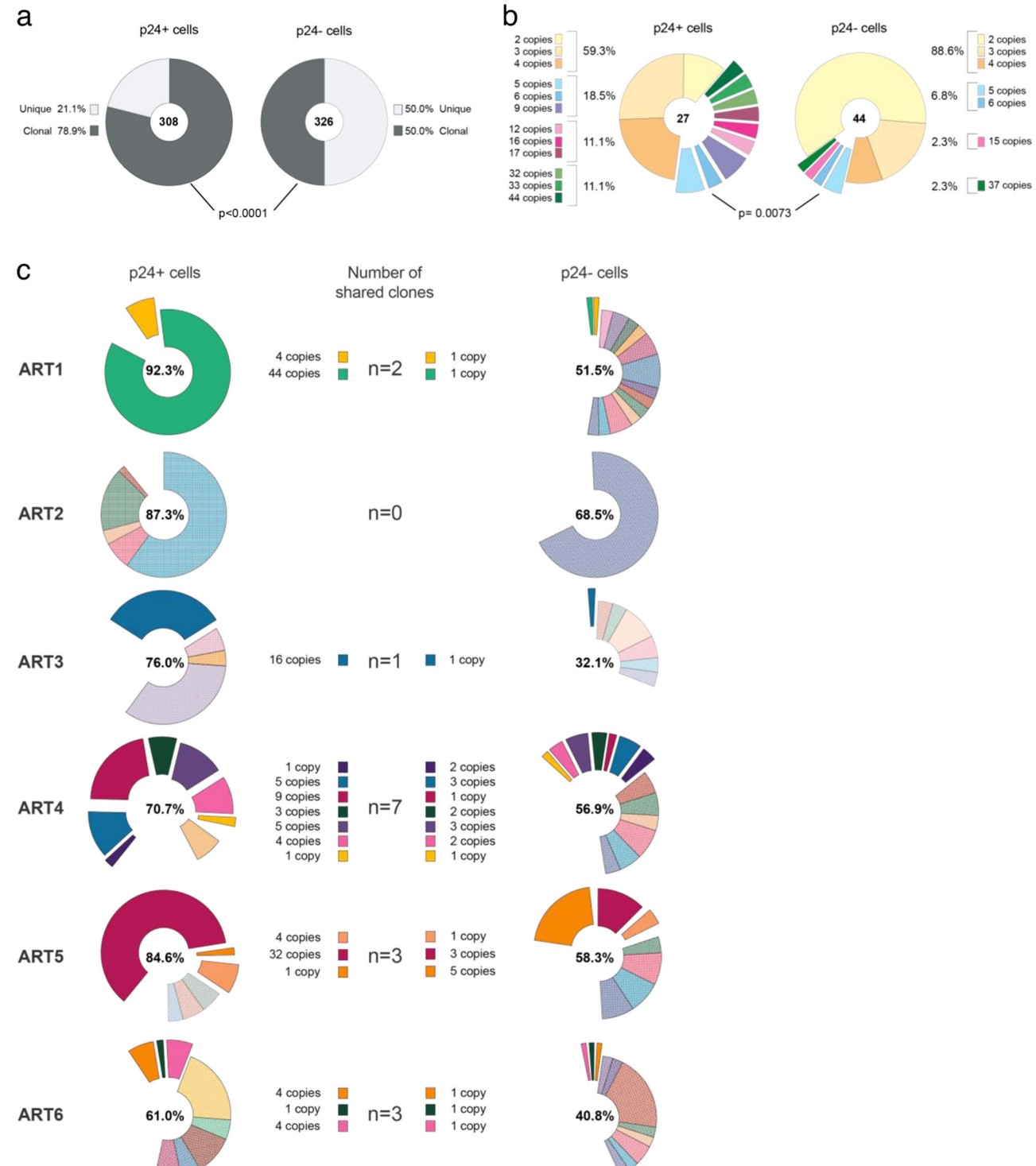

**Fig. 3 | Clonality of the inducible and non-induced reservoirs. a** Pie charts representing the proportions of unique (light grey) and clonally expanded (i.e., ≥2 copies, dark grey) proviruses in the p24+ and p24− populations from the 6 participants. Two-sided Fisher's exact test ($p < 0.0001$) of the contribution of clonally expanded proviral sequences to the total number of HIV genomes was performed between the two populations. The numbers of HIV genomes are indicated in each pie chart. **b** Distributions of clonal expansions from the 6 participants. Proportions of clones according to their size are represented for each cell population, with the total number of clonal expansions indicated in the pie charts. Difference in the contribution of clones with ≥5 copies (exploded from the piecharts) between p24+ and p24− cells was tested with two-sided Fisher's exact test ($p = 0.0073$). **c** Clonally expanded proviruses in the p24+ and p24− populations of the 6 participants. Each expansion is depicted by a different color. Shared clones between the p24+ and p24− populations are exploded from the charts and are depicted in solid colors. Contributions of clonal expansions to each population are indicated in the pie charts. The number of shared clones and their copy numbers are indicated in the middle column.

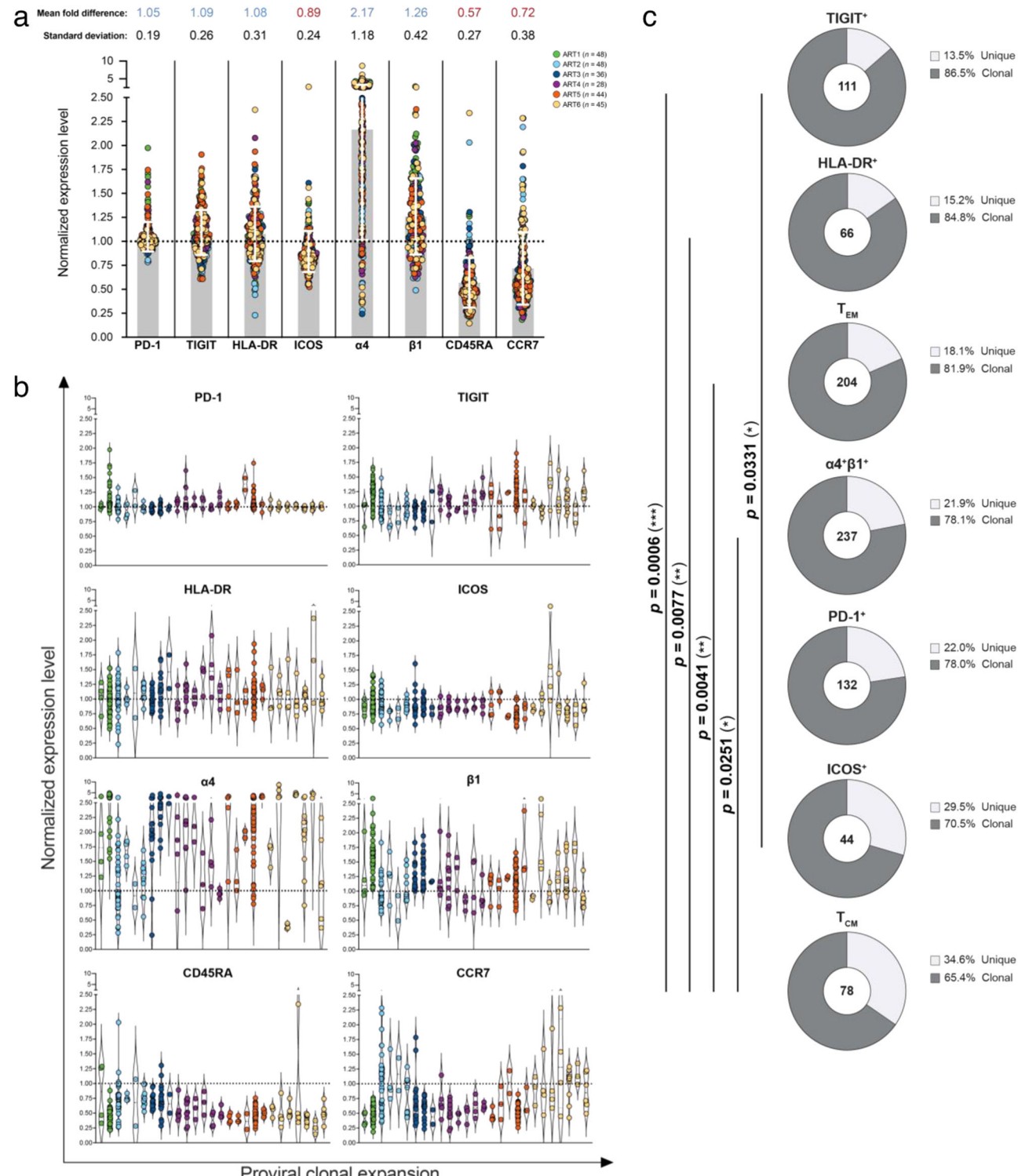

**Fig. 4 | Phenotype of clonally expanded cells harboring inducible and translation-competent proviruses.** Levels of expression of PD-1, TIGIT, HLA-DR, ICOS, α4, β1, CD45RA and CCR7 is represented by the ratio between the fluorescence intensity of a cellular marker on each p24+ single sorted-cell and the mean fluorescence intensity of this marker on all CD4+ T cells from the same participant. A normalized expression level above or below 1 (dotted line) reflects a higher or lower expression of this marker on a given p24+ cell compared to all CD4+ T cells, respectively. Each dot represents a single-sorted cell and is color-coded by the participant. **a** Normalized expression levels of each marker for all clonally expanded p24+ cells (n = 239 independent clonally expanded proviruses). Mean fold differences in levels of expression and standard deviations are indicated at the top of the graph. Grey bars indicate the mean normalized expression level for each marker. **b** Normalized expression levels of individual p24-expressing cells belonging to each individual clone. Single-sorted cells belonging to the same clone are grouped in each column and color-coded by the participant. **c** Pie charts representing the proportions of unique (light grey) and clonally expanded (dark grey) proviral sequences expressing each marker. The numbers of proviral sequences retrieved from p24+ cells expressing each marker are indicated in the center of the pie chart. Differences in the frequency of clonal proviruses between each subset were assessed by a two-sided Chi-square t-test (*p < 0.05; **p < 0.01; ***p < 0.001).

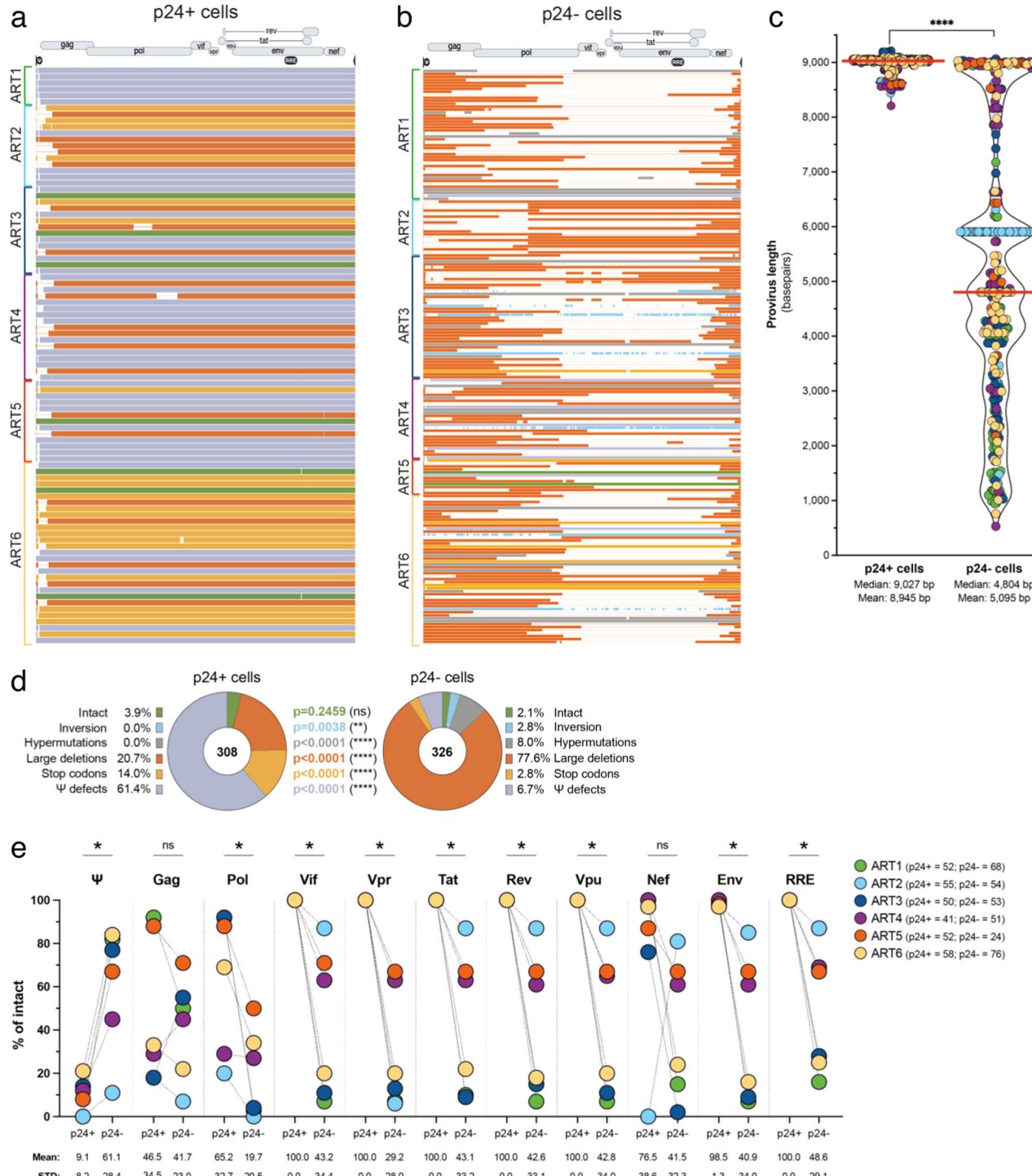

**Fig. 5 | Near full-length proviral sequencing and integrity analysis of HIV genomes retrieved from p24+ and p24− cells.** HIV genomes obtained from p24+ cells (**a**) and p24− cells (**b**) from the 6 participants were aligned to the annotated HXB2 reference sequence. Each sequence is color-coded according to its integrity category (see legend in 5d). Clonally expanded proviruses are only displayed once. **c** Lengths of individual HIV genomes retrieved from p24+ and p24− cells are plotted: Each dot corresponds to a proviral sequence and is color-coded by participant (see legend in 5e). The median of each violin plot is represented by a red horizontal line, and the mean and median values are indicated at the bottom of the graph. Difference in the length of HIV genomes between p24+ and p24− populations was assessed by the Kolmogorov-Smirnov test (****$p < 0.0001$). **d** Proportions

of proviruses displaying different types of genetic defects in p24+ and p24− cells. Numbers of proviral sequences analyzed are indicated in the pie charts. Differences in the proportion of proviruses displaying different types of defects between p24+ and p24− cells were assessed by the two-sided Fisher's exact test (**$p = 0.0038$; ****$p < 0.0001$). **e** The proportions of viral genomes with intact Ψ, gag, pol, vif, vpr, tat, rev, vpu, nef env and RRE (rev responsive element) were compared between p24+ and p24− cells. Each participant is color-coded, and total number of sequences analyzed are indicated next to the legend. Means and standard deviations (STD) are indicated at the bottom of the graph. Differences in the percentage of intact Ψ between p24+ and p24− populations were assessed by the nonparametric two-tailed Wilcoxon paired $t$-test (*$p = 0.0312$).

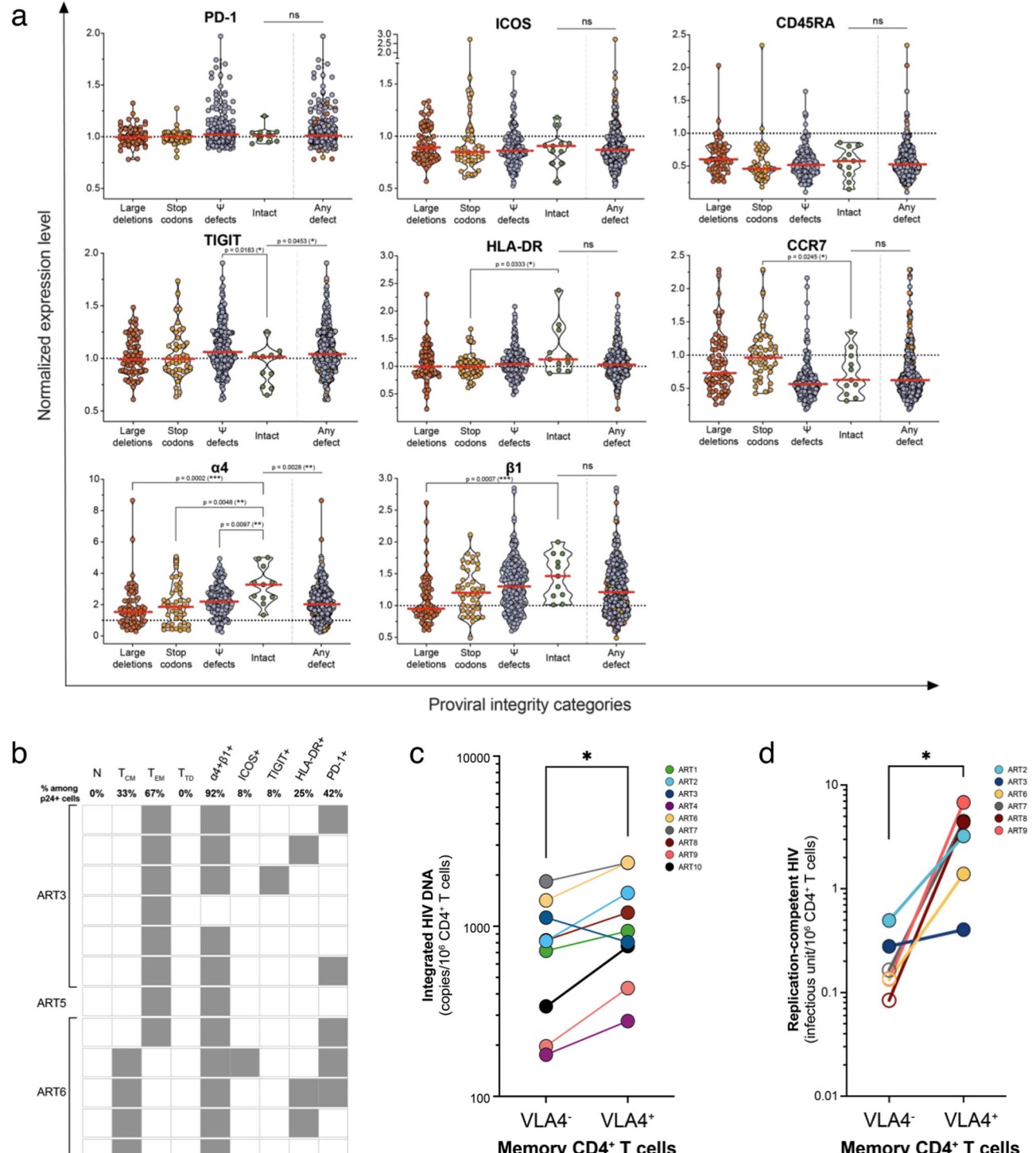

Fig. 6 | Phenotype of cells harboring inducible, translation- and replication-competent proviruses. a Normalized expression levels of PD-1, ICOS, CD45RA, TIGIT, HLA-DR, CCR7, α4, and β1 on cells harboring HIV genomes with different types of genetic defects. Each dot corresponds to a single cell, and cells are grouped according to the genetic defects found in the viral genomes they carry. In each panel, the violin plot on the right includes all defective proviruses grouped together. Two-tailed Mann-Whitney unpaired nonparametric t-tests were performed to assess differences in the expression level of each marker between intact proviruses and all other categories (*p < 0.05; **p < 0.01; ***p < 0.001). In all panels, the threshold ratio of 1 is indicated by a dotted line and the medians of the violin plots are represented by red lines. b Phenotypic analysis of the p24+ cells harboring intact HIV genomes. Grey square denotes the expression of a given marker.

Frequencies of p24+ cells with intact provirus expressing each marker are indicated at the top of the table. c Integrated HIV DNA (copies per million CD4+ T cells) was measured in sorted memory CD4+ T cells expressing or not VLA-4 from 9 ART-treated participants. Differences were assessed by the non-parametric two-tailed Wilcoxon paired t-test (*p = 0.0273). d Frequency of memory CD4+ T cells with replication-competent HIV was measured by QVOA in sorted cells from 6 participants expressing or not VLA-4. Differences in infectious units per million cells (IUPM) between memory CD4+ T cells expressing VLA-4 or not were assessed by the non-parametric two-tailed Wilcoxon paired t-test (*p = 0.0312). Samples with undetectable IUPM values were plotted at the limit of detection of the assay (open circles).

levels at the surface of CD4+ T cells harboring intact HIV genomes compared to cells carrying genomes with any genetic defect ($p < 0.01$ for all categories). The integrin β1 (CD29), which combines with α4 to make the lymphocyte homing receptor α4β1 (or very late antigen-4, VLA-4), was expressed at higher levels on CD4+ T cells harboring intact HIV genomes when compared to cells carrying genomes with large deletions ($p = 0.0007$). More importantly, all cells harboring intact proviruses co-expressed high levels of α4β1 (VLA-4), whereas only 39.7% of cells with large deletions did so (Fig. S4b).

In an attempt to identify common phenotypic features of HIV reservoir cells harboring genetically intact HIV genomes, we analyzed the phenotype of each p24+ cell individually ($n = 12$) (Fig. 6b). All cells with intact and inducible HIV genomes displayed a memory phenotype (33% $T_{CM}$ and 67% $T_{EM}$ cells). A minority of the cells harboring genetically intact genomes expressed ICOS, TIGIT or HLA-DR (8%, 8% and 25%, respectively), whereas PD-1 was more frequently expressed (42%). Strikingly, the vast majority (92%) of cells with intact genomes expressed α4 and β1, independently of their memory phenotype. Taken together, our results indicate that genetically intact and inducible HIV genomes primarily persist in memory CD4+ T cells expressing the adhesion molecule VLA-4.

## α4β1 enriches in CD4+ T cells with intact and replication-competent proviruses

Having demonstrated that genetically intact and inducible HIV genomes primarily persist in memory CD4+ T cells expressing higher levels of the integrin dimer α4β1, we sought to determine if this marker identifies cells harboring replication-competent HIV. We cell-sorted memory CD4+ T cells expressing VLA-4 (VLA-4+) or not (VLA-4-) from 9 virally suppressed PLWH (Fig. S5a). On average, VLA-4 was expressed by 23.7% CD4+ T cells and 36.6% memory CD4+ T cells (Fig. S5b). Compared to all CD4+ T cells, VLA-4+ cells were enriched in memory CD4+ T cells (both $T_{CM}$ and $T_{EM}$), expressed immune checkpoint molecules more frequently (PD-1, TIGIT), and expressed similar levels of immune activation markers (ICOS and HLA-DR) (Fig. S5c). We first measured the frequency of cells harboring integrated HIV DNA in sorted memory CD4+ T cells expressing VLA-4 or not. Memory CD4+ T cells expressing VLA-4 harbored slightly higher levels of integrated HIV DNA when compared to their VLA-4- counterparts (median of 936 and 820 copies/$10^6$ cells, respectively, $p = 0.027$) (Fig. 6c). We then measured the frequency of cells with replication-competent HIV using a modified quantitative viral outgrowth assay[74]. Large numbers of memory CD4+ T cells expressing VLA-4 or not (median of $6.2 \times 10^6$ and $6.7 \times 10^6$ cells, respectively) were cell-sorted from 6 virally suppressed PLWH, plated in limiting dilution, stimulated with CD3/CD28 antibodies and cultured for 3 weeks. In samples from all 6 participants, memory VLA-4+ cells harbored significantly higher frequencies of cells with replication-competent HIV than their VLA-4- counterparts (mean of 3.24 IUPM and 0.12 IUPM, respectively, $p = 0.031$) (Fig. 6d). Of note, in samples from 4 of the 6 participants, replication-competent HIV was exclusively retrieved from memory VLA-4+ cells and not from VLA-4- cells.

Altogether, these results indicate that VLA-4 expression not only identifies cells with intact inducible and translation-competent persistent proviruses but also cells carrying replication-competent HIV.

## Discussion

Single HIV genome approaches have provided extremely valuable information on the chromosomal location of proviral integration sites[8,21,24,34–37,68,75], on the T cell receptor repertoire and antigen specificity of HIV-infected cells[31,33–35,76] as well as on the integrity and clonality of viral genomes[8,25,26,29,30,36,37,49,59,62,68]. Nonetheless, these studies did not report on the phenotype of the cells that carry HIV proviruses and particularly those with genetically intact genomes. In this study, we developed an approach to analyze concomitantly the viral

genotype and cellular phenotype of cells carrying inducible and translation-competent HIV genomes.

Our approach relies on the combination of single-cell sorting of p24+ cells followed by near full-length HIV genome amplification. NFL genome sequencing approaches using long-distance PCRs have been suspected to underestimate the intact reservoir, since amplification of short, deleted proviruses may be favored over putatively intact 9 kb genomes[77,78]. Using single cells, we precisely measured the efficiency of our optimized method, which was consistently greater than 70%. This approach is, to our knowledge, the only one allowing association between the genetic integrity of inducible HIV genomes and the phenotype of the cells in which it persists.

Using this optimized approach, we observed a significantly higher proportion of clonally expanded genomes in p24+ cells compared to the pool of cells harboring non-induced and/or translation-incompetent proviruses (79% versus 50%, respectively). Although the proportion of clonally expanded genomes is likely underestimated in both populations due to the limited sampling, similar numbers of proviruses were analyzed in each compartment (308 and 326 genomes from p24+ and p24− cells, respectively), making unlikely that this observation results from a sampling bias. While it is now well-established that common antigens such CMV, influenza virus, EBV and *Mycobacterium tuberculosis* can induce proliferation of infected T cell clones[28,31,34,35], the higher clonality of infected cells with inducible and translation-competent proviruses is intriguing. It is possible that intact genomes provide the infected cells with a selective advantage to escape immune-mediated killing during proliferation. In support of this model, it was recently shown that functional Nef plays a role in the persistence of genetically intact HIV[44] and that Nef-specific CD8+ T cells are associated with the frequencies of infected cells[79]. Alternatively, the higher clonality of the inducible and translation-competent viral reservoir may reflect the lower inducibility of HIV genomes in $T_{CM}$ cells, a subset previously shown to be more refractory to reactivation[80] and which is characterized by a greater clonotypic diversity[76].

A striking observation was the extensive phenotypic diversity of CD4+ T cells harboring the exact same HIV genome, indicating that clonally expanded infected cells can display very different phenotypes. This observation is in line with the recent work by Weymar et al.[81] demonstrating that cells harboring clonally expanded intact genomes display diverse gene expression profiles. These findings suggest that infected CD4+ T cells undergoing clonal expansions can acquire or lose specific cell surface markers and that this process likely contributes to the phenotypic diversification of the reservoir. Cho et al. recently reported a concomitant increase in clonality and a decrease in the proviral diversity of the viral reservoir with time on ART[29]. Our results add another layer of information to these observations since they suggest that although the genetic complexity of the proviruses decreases with time on suppressive ART, their phenotypic diversity is likely to increase, a hypothesis that could be formally tested in a longitudinal study.

By comparing the viral integrity landscape of the inducible and translation-competent reservoir with its p24− counterpart, we observed that cells expressing viral capsid upon stimulation harbor longer, less deleted proviruses. However, genetically intact genomes were equally rare in both populations (<4%), and translation-competent proviruses often presented defects in the Ψ region or lacked the initial start codon in *gag*, as recently observed by Cole et al.[35]. Interestingly, neither of these defects interferes with the transcription and translation of the *gag* unspliced mRNA, and cells carrying these HIV genomes retained the capacity to express the viral capsid. Whether these Ψ locus aberrations could abrogate splicing or transcription of other HIV genes would require further investigations and a better understanding of the functional plasticity of this key region[26,29,82]. In addition, the high proportion of Ψ locus defects in

translation-competent genomes (61%) suggests that these may be positively selected during ART through a mechanism that remains to be identified.

By analyzing the phenotype of cells harboring intact and inducible viral genomes, we confirmed that both central and effector memory cells contribute to the persistence of HIV during ART[18,25,44,50]. Among the 8 cellular markers we analyzed, the integrins α4 and β1 were expressed at higher levels on the surface of CD4+ T cells harboring intact, inducible, and translation-competent genomes compared to cells carrying various genetic defects. However, as reported by others[27,35], p24+ cells were surprisingly not enriched in intact proviruses, which limited the number of intact genomes we could analyze. To overcome this limiting step, we sought to validate our intriguing phenotypic finding using QVOA, the gold standard assay to measure the inducible and replication-competent HIV reservoir. This complementary approach demonstrated a 27-fold enrichment in replication-competent HIV in CD4+ T cells expressing α4β1 compared to their negative counterparts. Importantly, α4β1 expression enriched for intact and replication-competent proviruses in all samples tested and may represent an interesting addition to previously described flow cytometry panels aimed at isolating HIV-infected cells[83]. The combination of α4 and β1 integrins, also known as the adhesion molecule VLA-4, is involved in trafficking of immune cells to inflammatory sites as well as in T cell co-stimulation[84]. It was recently demonstrated that expression of VLA-4 on resting CD4+ T cells favors their susceptibility to HIV infection in vitro[85] suggesting that VLA-4 may contribute to the reseeding of the reservoir by promoting the migration of resting CD4+ T cells to inflammatory tissues and facilitating their de novo infection during ART[86,87]. Although this model is attractive, results from many studies strongly suggest that reservoir replenishment is limited during ART and that the bulk of latently infected cells is established around the time of ART initiation[88]. In this context, the selective advantage that VLA-4 may confer to CD4+ T cells harboring intact and inducible genomes remains elusive. VLA-4 interaction with its ligand VCAM-1 can be blocked by the monoclonal antibody natalizumab, which is used to treat multiple sclerosis but has been associated with the development of progressive multifocal leukoencephalopathy (PML)[89]. Although natalizumab is unlikely to be a safe therapeutic option to target HIV reservoir cells in humans, novel generations of VLA-4 antagonists such as the recently approved α4 antagonist carotegrast methyl[90] could be tested in animal models.

In conclusion, our results indicate that clonal expansions in the reservoir lead to the phenotypic diversification of the cells harboring HIV genomes. Despite this diversity, the expression of VLA-4 appears to be maintained, suggesting that this adhesion molecule facilitates the persistence of latently infected cells harboring intact and replication-competent proviruses. Future studies will be needed to unravel the mechanism by which VLA-4 exerts this effect.

## Methods
### Study design
The study was approved by the McGill University Health Centre and the Centre Hospitalier de l'Université de Montréal review boards. All participants were adults and signed informed consent forms. All participants were male living with HIV under suppressive ART. They underwent leukapheresis to collect large numbers of PBMCs. Participant's characteristics are summarized in Table 1. PBMCs were isolated by Ficoll density gradient centrifugation and were cryopreserved in liquid nitrogen. CD4+ T cells were enriched by negative magnetic selection using the EasySep Human CD4 T Cell Enrichment Kit (StemCell Technology, 19052).

### HIV reservoir quantifications
Total HIV DNA (LTR/*gag*) and integrated HIV DNA (LTR/Alu) levels in CD4+ T cells were measured by real-time nested qPCR, as previously described[91]. Quantifications were performed in triplicates. Results were calculated based on a serial dilution of ACH-2 cells and expressed as HIV copies per million cells. Study participants were selected based on the frequency of p24+ cells measured by HIV-Flow (median 8 p24+ cells per million CD4+ T cells).

### HIV-flow staining
Enriched CD4+ T cells were processed for HIV-Flow assay[50]. Briefly, after 1 h pre-incubation with 5 μg/ml Brefeldin A (BFA) and 24 h stimulation with of 162 nM PMA and 1 μg/ml ionomycin in the presence of ARVs (200 nM lamivudine and 200 nM raltegravir), extracellular staining was performed using the following antibodies: Live/Dead Aqua Cell Stain (ThermoFisher Scientific cat.L34957), CD45RA APC-H7 (clone HI100; BD cat.560674), CCR7 BB700 (clone 3D12; BD cat.566437), PD-1 BV605 (clone EH12.2H7; Biolegend cat.329924), TIGIT eF450 (clone MBSA43; eBioscience cat.48-9500-42), HLA-DR AlexaFluor700 (clone G46-6; BD cat.560743), ICOS BV785 (clone C398.4 A; Biolegend cat.313534), α4/CD49d PE-Cy7 (clone 9F10; Biolegend cat.304313) and β1/CD29 BB515 (clone MAR4; BD cat.564565). Cells were simultaneously fixed and permeabilized with the FoxP3 Buffer Set (eBioscience), followed by intracellular staining of HIV p24 with clone 28B7 APC (MediMabs cat.MM-0289-APC) and clone KC57 PE (Beckman Coulter cat.6604667). All samples were resuspended at a final concentration of $1 \times 10^6$ cells/ml in PBS and filtered prior to cell sorting.

### Cell sorting of p24− and p24+cells
Viable p24 simple positive cells (28B7+ or KC57+ only) and double negative (28B7⁻KC57⁻) were bulk sorted prior to single-cell sorting (BD FACS ARIA III). 0.25 to 1 million p24− cells were sorted and resuspended in 100 μL of DirectPCR Lysis Reagent (Viagen Biotech cat.301-C) containing 0.4 mg/mL Proteinase K (Wisent cat.25530−015). Cells were incubated at 55 °C for 16 h for cell lysis followed by 15 min at 85 °C to inactivate proteinase K. Lysates were kept at −20 °C until NFL amplification. Viable p24 double positive cells (28B7+KC57+) were index single-cell sorted (BD FACS ARIA III) in 12-wells PCR strips (Fisher cat.AB1114) containing 8 μL of DirectPCR Lysis Reagent (Viagen Biotech cat.301-C) and 0.4 mg/mL Proteinase K (Wisent cat.25530−015). The ACH-2 cell line (NIH HIV Reagent Program cat. ARP-349) was single-sorted to obtain positive control cells. Strips were incubated at 55 °C for 1 h for cell lysis followed by 15 min at 85 °C to inactivate proteinase K. Index-sorting data of p24+ cells were analyzed using FlowJo version 10.5.3, using the indexed sorting script (https://flowjo. typepad.com/the_daily_dongle/2016/12/how-to-use-flowjos-script-editor-with-index-sorted-data.html). To compare the phenotype of single-sorted cells from different participants and from multiple sorts, we calculated the ratio between the fluorescence intensity (FI) of each fluorochrome on a given cell by the mean fluorescence intensity (MFI) of all CD4+ T cells from the same participant.

### Single-amplicon NFL nested PCR amplification
This single-amplicon near full-length HIV nested PCR is derived from previous assays[25,62]. For bulk sorted cells, lysates were first diluted to reach a concentration of 1.5−3 copies of integrated HIV DNA per well, and 12 PCR reactions of each dilution were performed to determine the proper dilution to achieve 33% of positive wells. For single-sorted cells, PCR mix was added to each well immediately after lysis. HIV proviruses were pre-amplified in a 25-cycles 3-steps PCR reaction in a total volume of 40 μL containing Invitrogen Platinum SuperFi II MasterMix (ThermoFisher cat.12358050) with 0.2 μM of each primer (638 F: 5'-GCGCCCGAACAGGGACYTGAAARCGAAAG-3'; BLOuterR: 5'-TGAGGG ATCTCTAGTTACCAGAGTC-3'). 5 μL of each 4X diluted pre-amplification product were used in a second round of amplification, using inner primers (263 F: 5-'GACCTGAAAGCGAAAGGGAAAC-3'; 280 R: 5'CTAGTTACCAGAGTCACACAACAGACG-3') for 30 cycles in a 30 μL final volume reaction. ACH-2 single-sorted cell and diluted

extracted DNA were used as positive controls for sorting and PCR amplification. Positive reactions were determined by visualization of amplicons on a 0.8% agarose gel.

## PacBio sequencing

All amplicons were sequenced using the PacBio next-generation sequencing platform. Each HIV pre-amplified DNA were re-amplified with barcoded inner PCR primers (96 different PacBio barcodes). Barcoded amplicons were purified using AMPure XP beads (Beckman Coulter cat.A63881), following manufacturer's instructions, prior to Nanodrop quantification. 50 ng of each of the 96-barcoded amplicons were pooled together and sequenced on a Sequel or Sequel II instrument (Génome Québec, Canada; DNA Link, South Korea).

## PacBio demultiplex and cleaning

PacBio demultiplexing procedure was previously described[27]. The demultiplex barcodes analysis was powered by the Lima PacBio software v2.0.0. High-quality phased consensus sequences representing near full HIV-1 genome sequences with high fidelity and without reconstruction were generated with the LAA PacBio algorithm v2.4.2. Any sequences that did not blast with HIV, that blasted with ACH-2 positive control, or that lacked one of the primer sequences were discarded from further analysis. A minimum of 10 reads per sequence were used as a coverage threshold.

## Integrity analysis

All proviral sequences were aligned to the reference HXB2 sequence using Multiple Fast Fourier transform algorithm (MAFFT) with strategy E-INS-i and a scoring matrix of 1PAM/k = 2 (online https://mafft.cbrc.jp/alignment/server/ or with Geneious Prime (v2021.1.1) plugging). For integrity analysis, all sequences containing (in this order) inversions, hypermutations, large internal deletions, stop codons and/or frameshift, defects in the Ψ locus or small internal deletions in a coding region, were considered defective. Inversions were detected manually at the alignment step, since only the reverse complement of inversions can be aligned with HXB2. Hypermutations were detected using the online HIV Database QC Tool (https://www.hiv.lanl.gov/content/sequence/QC/index.html). Genomes with large internal deletions were defined as any provirus shorter than 8.800 bp (excluding the primer's regions). Stop codons/frameshifts and small internal deletions (>5% of ORF expected length) were identified using both the HIV Database Gene Cutter online tool (https://www.hiv.lanl.gov/content/sequence/GENE_CUTTER/cutter.html) and Proseq IT (https://psd.cancer.gov/tools/pvs_annot.php). Any defect in the Ψ locus (MSD point mutation; stem loop 2 deletion; packaging signal deletion) were determined using the Proseq IT tool, and manually confirmed by looking at the alignment with the reference HXB2 sequence. The intactness 9 coding regions (*gag, pol, vif, vpr, tat, rev, vpu, nef, env*) and 2 regulatory regions (Ψ and RRE) were evaluated by assessing the presence of a start codon (except for pol), and the lack of internal stop codons, frameshift or deletion over 5% of the ORF's length.

## Clonality analysis

Clonal sequences were defined as proviral amplicons 100% identical to each other, and were determined using the HIV Database Elim Dupes online tool (https://www.hiv.lanl.gov/content/sequence/elimdupesv2/elimdupes.html), and confirmed using the Geneious Prime diversity tool. Phylogenic trees were built with IQ-Tree2, using a Maximum-likelihood tree GTR + I + G model, 1,000 bootstraps.

## Modified quantitative Viral Outgrowth Assay (mQVOA) in sorted VLA-4 population

CD4+ T cells were obtained by negative magnetic selection from $300 \times 10^6$ PBMCs and stained with the following antibodies: Live/Dead Aqua Cell Stain (ThermoFisher Scientific cat.L34957), CD8 PB (clone RPA-T8; BD cat.558207), CD14 V450 (clone MΦP9; BD cat.560349), CD45RA APC-H7 (clone HI100; BD cat.560674), α4/CD49d PE-Cy7 (clone 9F10; Biolegend cat.304313) and β1/CD29 BB515 (clone MAR4; BD cat.564565). Viable memory CD4+ T cells expressing VLA-4 (CD8-/CD14- CD45RA- α4$^{high}$ β1$^{high}$) or not (CD8-/CD14- CD45RA- α4$^{low/-}$ β1$^{low/-}$) were sorted in 5 mL FACS tubes. Cells were rested for 2 h at a final concentration of 1.5 million per mL, prior to serial dilution in culture plates (Costar) coated with 2.5 µg/ml anti-CD3 (clone OKT3) and 1 µg/ml anti-CD28 (clone CD28.2) antibodies as described elsewhere[50]. MOLT-4 CCR5 + target cells (NIH HIV Reagent Program cat. ARP-4984) were added 2 days post-sort at a final concentration of $0.5 \times 10^6$ cells/mL and the culture was maintained for 21-days. Supernatants were collected at day 7, 11, 14, 18, and 21 for soluble HIV-p24 protein quantification by ELISA[92]. Infectious units per million of cells (IUPM) were determined for each population based on the number of positive wells for soluble p24 protein (http://silicianolab.johnshopkins.edu/)[93].

## Statistical analyses

All data were analyzed using GraphPad Prism v9.3.0. Statistical tests are indicated in the figure legends.

## Reporting summary

Further information on research design is available in the Nature Portfolio Reporting Summary linked to this article.

# Data availability

The sequences reported in this manuscript are available in GenBank with the following reference codes, ON816029.1 to ON816663.1 (https://www.ncbi.nlm.nih.gov/popset/2306699925). The datasets generated in this study have been deposited as a Source Data file in the Figshare database and can be accessed here: https://doi.org/10.6084/m9.figshare.21912840. Source data are provided with this paper.

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

## Acknowledgements

The authors are grateful to the people living with HIV who volunteered to participate in this study. We thank Josée Girouard, Mario Legault, and Claude Gagné for recruitment, coordination, and clinical assistance with study participants. We thank Bonnie Hiener and Sarah Palmer for their precious help with the FLIPS assay and with the analysis pipeline. We thank the flow cytometry core at the CRCHUM, managed by Dominique Gauchat and Philippe St-Onge for cell sorting as well as the NC3 core (Olfa Debbeche). The following reagents were obtained through the NIH HIV Reagent Program, Division of AIDS, NIAID, NIH: ACH-2 Cells, ARP-349, contributed by Dr. Thomas Folks, and ARP-4984, contributed by Dr. Masanori Baba, Dr. Hiroshi Miyake, and Dr. Yuji Iizaw. This work was supported by the Canadian Institutes for Health Research (CIHR; operating grants #148540 and #451304 and the Canadian HIV Cure Enterprise

(CanCURE) Team Grant HB2 – 164064), the National Institute of Allergy and Infectious Diseases, National Institute for Drug Abuse, National Institute of Neurological Disorders and Stroke, National Institute of Diabetes and Digestive and Kidney Diseases and the National Heart, Lung, and Blood Institute [NIAID/NIDA/NINDS/NIDDK/NHLBI, grant number UM1AI164560: Delaney AIDS Research Enterprise (DARE) to Cure HIV], the Réseau SIDA et maladies infectieuses du Fonds de Recherche du Québec - Santé (FRQ-S). C.D. is supported by a doctoral fellowship from the CIHR (#413313) and from the FRQ-S (#275429). J.-P.R. is the holder of the Louis Lowenstein Chair in Hematology and Oncology, McGill University. N.C. is supported by Research Scholar Career Awards of the FRQ-S (#253292). The funders had no role in study design, data collection and analysis, decision to publish, or preparation of the manuscript.

## Author contributions

C.D. designed and performed the experiments, analyzed the data, and wrote the manuscript draft. C.R. B.R., and B.M. performed the bioinformatics analyses of the sequencing data. R.T. and J.-P.R. managed the recruitment of the participants. A.A. performed part of the experiments. M.P. and M.M. provided conceptual advice. R.F. designed and performed part experiments, provided conceptual advice. N.C. designed the experiments, analyzed the data, and wrote the manuscript draft. All authors read, edited and approved the manuscript.

## Competing interests

The authors declare no competing interests.
