## [Peer Review File · Nature Communications]

Phenotypic characterization of single CD4+ T cells harboring genetically intact and inducible HIV genomesReviewer #1 (Remarks to the Author):

Dufour and colleagues report on experiments designed to elucidate the phenotype of HIV-1 infected latent cells. To do so they use a novel flow based approach in which inducible p24 expressing cells are identified and purified and their nucleic acids subjected to near full length sequence analysis. Proviral amplification is by a modification of reported methods and sequencing is by Pacbio. In addition to p24 staining they also use 8 additional cell surface markers to document the phenotype of the sorted cells. 6 individuals were studied to obtain 309 near full length sequences from the p24+ cells only 3.9% of which were intact, i.e. only 12 inducible sequences were intact. 326 sequences were obtained from p24- cells of which 2.1% were intact i.e. only 7 intact proviruses. So most of what they are looking at are inducible or non-inducible near full length proviruses that are defective and do not contribute to rebound.

The key findings are: 1. that both inducible and non-inducible viruses are clonal with a greater proportion of clonal viruses in the p24+ compartment; 2. Cells belonging to the same expanded clone are found in different T cell subsets as defined by the flow markers; 3. Large deletions are more common in the defective proviruses found in the p24- cells; p24+ cells express higher levels of the VLA-4.

The strengths of this manuscript is the new method, and the weakness is the paucity of intact sequences on which the conclusions are based. In addition, some of the conclusions are not novel including the finding that clones of T cells that harbor intact viruses have diverse phenotypes DOI:<https://doi.org/10.1016/j.celrep.2022.111311>. Despite these shortcomings this is a valuable addition to the literature that would benefit from additional clarity on the limitations of the study in terms of the numbers of sequences analyzed, the absolute number of intact sequences being evaluated, citation and discussion of the literature relevant to analysis of the phenotype of cells found in expanded intact clones.

Reviewer #2 (Remarks to the Author):

In their previous manuscript (Pardons et al. PloS Pathog. 2019), the authors reported that blood CD4 T cell isolated from untreated and cART-treated HIV-1 infected individual expressing p24 (without activation and upon activation respectively) are phenotypically diverse and enriched for the expression of VLA-4 ($\alpha 4/\beta 1$). In the present manuscript, Dufour et al. extend their observation by characterizing the clonality and the genetic integrity of the provirus contained in different subpopulation of CD4 T cell including those expressing $\alpha 4/\beta 1$. They evidence: 1) the clonality of the inducible and translational competent proviruses. 2) Phenotypically different CD4 T cells harbor the same clonally expanded provirus. 3) distinct genomic defect within the proviruses in p24+ and p24- cells. 4) VLA4 is highly expressed by p24+ cells with intact proviral genome. 5) VLA4+ CD4 T cells are highly enriched in replication competent provirus as compared to VLA4 minus CD4 T cells. The topic is of importance, the experiments are well controlled and, in most case, support the authors' conclusions. However, with the exception of the outstanding results shown in figure 6 (particularly panels C and D), the manuscript lacks novelty. Indeed, most of the findings were somehow reported (Sannier et al., 2021, Cell Reports 36, 109643, Pardons et al. PloS Pathog. 2019). Given the importance of the results shown in figure 6, it is frustrating to not see a more in-depth analyses of the CD4 T cell expressing $\alpha 4/\beta 1$. Combining their excellent experimental scheme described in figure 1A with the FLOW-FISH reported in Sannier et al. 2021, would have been of great importance.

Major comments:

*Figure 4A and 4B: Analyses of double positive $\alpha 4/\beta 1$ cell need to be shown.

*Figure 4C: Are the observed differences statistically significant?

*Figure 5:

- The finding that p24+ and p24- cells harbor proviruses with distinct genetic defects questions the use of "non-inducible" which usually designates transcriptionally non-inducible proviruses. Indeed, 91.8% of the proviruses contained in p24- cells are incompetent in producing p24 even if they were transcriptionally induced (Sannier et al., 2021, Cell Reports 36, 109643). Multiple rounds of activation will be important to determine what is the proportion of the proviruses contained in the p24 neg cells for which the limiting step is transcription versus translation.

-5A: How the authors explain the expression of p24 from some of the proviruses with large deletion and Stop codons? Did the authors analyze the gag open reading frame of these proviruses? Do they have an ATG? One of these proviruses has lost the entire 5' region to the middle of the pol gene.

-The statement "Interestingly, these defects corresponded to point mutations in the MSD site (8.5% in p24+), deletion of stem loop 2 (34.9% in p24+) or larger deletion of the Ψ locus (56.6% in p24+) which should theoretically preclude transcription, splicing and translation" is incorrect. Indeed, Ψ has no function in transcription and translation. The major splice donor site (MSD) play no role in gag expression, since gag is expressed from unspliced viral mRNA.

-Similarly, the authors state "Since these defects were found in cells expressing the viral protein p24, this suggests the existence of compensatory splicing mechanisms in these putatively defective genomes, as suggested previously". The major splice donor site (MSD) play no role in gag expression, since gag is expressed from unspliced viral mRNA.

*Figure 6A: The authors should show the analyses of double positive $\alpha 4/\beta 1$ CD4 T cells.

Response to the Reviewer's comments

Reviewer #1 (Remarks to the Author):

Dufour and colleagues report on experiments designed to elucidate the phenotype of HIV-1 infected latent cells. To do so they use a novel flow based approach in which inducible p24 expressing cells are identified and purified and their nucleic acids subjected to near full length sequence analysis. Proviral amplification is by a modification of reported methods and sequencing is by Pacbio. In addition to p24 staining they also use 8 additional cell surface markers to document the phenotype of the sorted cells. 6 individuals were studied to obtain 309 near full length sequences from the p24+ cells only 3.9% of which were intact, i.e. only 12 inducible sequences were intact. 326 sequences were obtained from p24- cells of which 2.1% were intact i.e. only 7 intact proviruses. So most of what they are looking at are inducible or non-inducible near full length proviruses that are defective and do not contribute to rebound.

The key findings are: 1. that both inducible and non-inducible viruses are clonal with a greater proportion of clonal viruses in the p24+ compartment; 2. Cells belonging to the same expanded clone are found in different T cell subsets as defined by the flow markers; 3. Large deletions are more common in the defective proviruses found in the p24- cells; p24+ cells express higher levels of the VLA-4.

The strengths of this manuscript is the new method, and the weakness is the paucity of intact sequences on which the conclusions are based.

We appreciate the Reviewer's evaluation of our manuscript.

We acknowledge that the total number of intact sequences is limited. Overall, 19 intact proviruses were identified among 635 sequences, which represents a frequency of 3%. However, this frequency and the total number of sequences are comparable to what was observed in several recent studies using similar sequencing approaches [Lee *et al.* JCI 2017 (4% of 621 sequences), Hiener *et al.* Cell Reports 2017 (5% of 531 sequences) and Horsburgh *et al.* AIDS 2020 (5% of 732 sequences)]. In addition, we would like to point out that our analysis pipeline is extremely stringent to qualify proviral intactness: no inversion, no hypermutations, no large internal deletion, no deleterious stop codon, frameshift, small internal deletion in any of the 9 ORFs except in *nef*, no defects in the 4 stem loops of the psi locus, no deletion of the RRE region.

We acknowledge that a larger number of sequences from p24+ cells may have strengthened our conclusions on the phenotype of cells harbouring intact genomes. Nonetheless, we confirmed our main finding (higher expression of VLA-4 on cells harboring intact proviruses) using a completely different approach, by directly assessing the replication competency of the proviruses persisting in VLA-4+ CD4+ T cells. Therefore, although our sequencing approach could be seen as exploratory, it was confirmed by measuring the inducibility and replication competency of the proviruses persisting in VLA-4+ cells. We strongly believe that this approach convincingly supports our findings even if our initial observation was made on a limited number of sequences.

In addition, some of the conclusions are not novel including the finding that clones of T cells that harbor intact viruses have diverse phenotypes DOI:<https://doi.org/10.1016/j.celrep.2022.111311>.

The Reviewer mentioned the study by Weymar *et al.* in which the authors analyzed the transcriptome and not the phenotype of cells harboring intact HIV genomes. We agree that this elegant study shows that clones of T cells with intact proviruses can display diverse gene expression profiles, which is reminiscent of our own findings. We would like to emphasize that our study is based on the expression of cell-surface markers measured by flow cytometry and represents, to our knowledge, the first study to report on the phenotype of HIV-infected cells harboring genetically intact HIV genomes. Our study provides data that complement and extend the work by Weymar *et al.* We modified the manuscript to cite and discuss the work of Weymar *et al.* (lines 316-320).

Despite these shortcomings this is a valuable addition to the literature that would benefit from additional clarity on the limitations of the study in terms of the numbers of sequences analyzed, the absolute number of intact sequences being evaluated, citation and discussion of the literature relevant to analysis of the phenotype of cells found in expanded intact clones.

We would like to thank the Reviewer for their positive evaluation of our manuscript. As recommended by the Reviewer, we extensively modified our manuscript to discuss additional studies that describe the phenotype of HIV-infected cells in people with HIV receiving ART (lines 66-76). We also acknowledged the relatively small number of intact HIV sequences analyzed (lines 344-352).

Reviewer #2 (Remarks to the Author):

In their previous manuscript (Pardons et al. PloS Pathog. 2019), the authors reported that blood CD4 T cell isolated from untreated and cART-treated HIV-1 infected individual expressing p24 (without activation and upon activation respectively) are phenotypically diverse and enriched for the expression of VLA-4 ($\alpha 4/\beta 1$). In the present manuscript, Dufour et al. extend their observation by characterizing the clonality and the genetic integrity of the provirus contained in different subpopulation of CD4 T cell including those expressing $\alpha 4/\beta 1$. They evidence: 1) the clonality of the inducible and translational competent proviruses. 2) Phenotypically different CD4 T cells harbor the same clonally expanded provirus. 3) distinct genomic defect within the proviruses in p24+ and p24- cells. 4) VLA4 is highly expressed by p24+ cells with intact proviral genome. 5) VLA4+ CD4 T cells are highly enriched in replication competent provirus as compared to VLA4 minus CD4 T cells.

The topic is of importance, the experiments are well controlled and, in most case, support the authors' conclusions. However, with the exception of the outstanding results shown in figure 6 (particularly panels C and D), the manuscript lacks novelty. Indeed, most of the findings were somehow reported (Sannier et al., 2021, Cell Reports 36, 109643, Pardons et al. PloS Pathog. 2019). Given the importance of the results shown in figure 6, it is frustrating to not see a more in-depth analyses of the CD4 T cell expressing $\alpha 4/\beta 1$. Combining their excellent experimental scheme described in figure 1A with the FLOW-FISH reported in Sannier et al. 2021, would have been of great importance.

We would like to thank the Reviewer for their careful evaluation of our manuscript and for highlighting the strengths of our study.

In response to the Reviewer's comment, we would like to better explain the novelty of our findings. To our knowledge, our study is the first to report on the phenotype of cells with genetically intact HIV genomes: the discovery that VLA-4 ($\alpha 4/\beta 1$) can be used as a marker for cells harboring replication-competent virus has not been reported before. In addition, we are not aware of a previous study reporting on the phenotypic diversity of infected clones nor on the higher clonality of p24+ cells compared to non-induced and/or translation-incompetent proviruses.

To address the reviewer's concern regarding the lack of more in-depth analysis of the CD4+ T cells expressing $\alpha 4/\beta 1$, we generated novel data on the phenotype of VLA-4+ ($\alpha 4/\beta 1$) CD4+ T cells from the same 6 participants (added to S5C, lines 257-260). Compared to all CD4+ T cells, VLA-4+ cells were enriched in memory CD4+ T cells (both T_{CM} and T_{EM}), expressed immune checkpoint molecules more frequently (PD-1, TIGIT), and expressed similar levels of immune activation markers (ICOS and HLA-DR).

Supplementary figure 5C. Frequencies of VLA-4 ($\alpha 4/\beta 1$) and CD4+ cells displaying a TCM or TEM phenotype and expressing PD-1, TIGIT, ICOS and HLA-DR in the blood of 6 ART-suppressed participants. Mean frequencies and standard deviations are indicated at the top of the graph.

Major comments:

*Figure 4A and 4B: Analyses of double positive $\alpha 4/\beta 1$ cell need to be shown.

We followed the Reviewer's suggestion and represented the normalized expression levels of $\alpha 4$ and $\beta 1$ in a single plot for every p24+ cells from Figure 4A. We also represented the normalized-expression levels of $\alpha 4\beta 1$ of each individual clone, as shown in Figure 4B. These new analyses are now included in the revised manuscript (now S3A and S3B).

Supplementary figure 3: $\alpha 4\beta 1$ (VLA-4) phenotype of clonally expanded cells harboring inducible and translation-competent proviruses. Levels of expression of $\alpha 4$ and $\beta 1$ are represented by the ratio between the fluorescence intensity of a cellular marker on each p24+ single sorted-cell and the mean fluorescence intensity of this marker on all CD4+ T cells from the same participant. A normalized expression level above or below 1 (dotted line) reflects a higher or lower expression of this marker on a given p24+ cell compared to all CD4+ T cells, respectively. Each dot represents a single-sorted cell and is color-coded by participant. **A.** Normalized double-expression levels of $\alpha 4\beta 1$ for all clonally expanded p24+ cells. Grey bars indicate the mean normalized expression level for each marker with its standard deviation in white. **B.** Normalized double-expression levels of $\alpha 4\beta 1$ of individual p24-expressing cells belonging to each individual clone.

*Figure 4C: Are the observed differences statistically significant?

We apologize for omitting statistical analyses of these data in our original manuscript. To address the Reviewer's concern, we performed chi-test to determine if the clonality index differ between the different subsets we analyzed. We found that T_{CM} cells harbour a significantly lower proportion of proviral clones compared to the TIGIT+ ($p=0.0006$), HLA-DR+ ($p=0.0077$), T_{EM} ($p=0.0041$), and $\alpha 4+\beta 1+$ ($p=0.251$), subsets. In addition, ICOS+ cells harboured lower proportion of clones than TIGIT+ cells ($p=0.0193$). We modified Figure 4C to represent these differences and modified the manuscript accordingly (lines 161-164).

Figure 4C. Pie charts representing the proportions of unique (light grey) and clonally expanded (dark grey) proviral sequences expressing each marker. Numbers of proviral sequences retrieved from p24+ cells expressing each marker are indicated in the center of the pie chart. Differences in frequency of clonal proviruses between each subset were assessed by Chi t-test (*: p<0.05; **: p<0.01; ***: p<0.001).

***Figure 5:**

- The finding that p24+ and p24- cells harbor proviruses with distinct genetic defect questions the use of “non-inducible” which usually designates transcriptionally non-inducible proviruses. Indeed, 91.8% of the proviruses contained in p24- cells are incompetent in producing p24 even if they were transcriptionally induced (Sannier et al., 2021, Cell Reports 36, 109643).

The reviewer raised an important point. Proviral genomes retrieved from p24- cells are located in cells that do not express the viral capsid but may have the capacity to do so. We cannot determine, using HIV-Flow, if the lack of p24 expression is due to blockade in transcriptional inducibility or in protein translation. However, since 16 identical proviral genomes were found in both p24+ and p24- compartments, this strongly suggests that a proportion of p24- cells are HIV translation-competent but failed to produce viral protein.

We agree with the Reviewer that RNA Flow FISH is well-suited to investigate the transcriptional profile of HIV-infected cells, which was the scope of a previous study (Sannier *et al.* Cell Rep 2021). In the current manuscript, we focused on cells that produce p24, since this readout is closer to the production of viral particles.

We acknowledge that the term “non-inducible” can refer to a lack of transcriptional activity and is confusing. We modified for “non-induced and/or translation incompetent” in the manuscript in order to better define infected p24- cells.

Multiple rounds of activation will be important to determine what is the proportion of the proviruses contained in the p24 neg cells for which the limiting step is transcription versus translation. As noted by the Reviewer, it is known that a single round of activation using PHA is not sufficient to reverse latency in every latently infected cells (Ho *et al.* Cell 2013). However, flow-cytometry based assays that stain for HIV protein (HIV-Flow; Pardons *et al.* PLoS Pathog 2019) or RNA (RNA-Flow FISH; Baxter *et al.* Cell Host Microbe 2016) require fixation and permeabilization steps, which kill the cells. Therefore, this approach does not allow us to isolate p24- cells after the first round of reactivation to perform additional rounds of stimulation. Future studies will be needed to determine if genetically intact genomes in the p24- fraction are deeply latent and refractory to reactivation.

-5A: How the authors explain the expression of p24 from some of the proviruses with large deletion and Stop codons? Did the authors analyzed the gag open reading frame of these proviruses? Do they have an ATG?

We thank the Reviewer for raising this point. Among the 94 unique proviral sequences retrieved from p24+ cells in our study, 36 lacked the initial start codon (position 790 in HXB2). However, with the exception of one provirus with complete deletion of the *gag* ORF (addressed in the next point from this Reviewer), we identified a second ATG located further in all these sequences which could substitute for this deletion (Figure below). Of note, 3 of these “alternative *gag* start” sequences have premature stop codons 9 nucleotides upstream of the original stop codon which should not impact p24 translation. This figure has been added to the revised manuscript (now S4A) and is described in the results section (lines 194-199).

Supplementary figure 4: Additional p24+ cells integrity analysis. A. Alternative ATG start codon in the gag ORF of 35 unique HIV sequences retrieved in p24+ cells from 6 ART-suppressed participants. Initial start codon (HXB2 position 790), p24 region (HXB2 position 1,186) and “later gag start codon” (HXB2 equivalent position) are indicated.

One of these proviruses has lost the entire 5' region to the middle of the pol gene. The Reviewer refers to a unique proviral genome that was identified among the p24+ cells and displayed a complete gag-pol deletion, as represented below (ART6_63H7_ID_53_4506_10 accession number ON816545 <https://ncbi.nlm.nih.gov/nuccore/ON816545.1>).

Despite this major 5' deletion, we found the sequences of both sequencing primers at the 5' and 3' ends. This sequence (identified by a red arrow in the phylogenetic tree below, for the Reviewer's consideration) clustered with other proviruses from this participant. The most likely explanation for this intriguing finding is an issue at the sequencing step. Using our quality control pipeline, we had no specific reason to exclude this sequence from our analysis and decided to keep it in our study. We discussed this specific proviral sequence in the revised manuscript (lines 200-202).

-The statement “Interestingly, these defects corresponded to point mutations in the MSD site (8.5% in p24+), deletion of stem loop 2 (34.9% in p24+) or larger deletion of the Ψ locus (56.6% in p24+) which should theoretically preclude transcription, splicing and translation” is incorrect. Indeed, Ψ has no function in transcription and translation. The major splice donor site (MSD) play no role in gag expression, since gag is expressed from unspliced viral mRNA.

We apologize for this error and thank the Reviewer for noticing it. We corrected the revised manuscript accordingly (lines 193-194).

-Similarly, the authors state “Since these defects were found in cells expressing the viral protein p24, this suggests the existence of compensatory splicing mechanisms in these putatively defective genomes, as suggested previously”. The major splice donor site (MSD) play no role in gag expression, since gag is expressed from unspliced viral mRNA.

We also corrected this error in the revised version of the manuscript (lines 193-194 and 331-336).

*Figure 6A: The authors should show the analyses of double positive $\alpha 4/\beta 1$ CD4 T cells.

We followed the reviewer’s recommendation and represented the double-normalized-expression of $\alpha 4\beta 1$ of each p24+ cell, based on their proviral integrity, as shown in Figure 6A. All cells with intact proviruses co-expressed high levels of $\alpha 4$ and $\beta 1$, which was not seen in cells harboring defective genomes.

This figure has been added as supplementary data (now S4B) (lines 237-239).

Supplementary figure 4: Additional p24+ cells integrity analysis. B. Normalized double-expression levels of $\alpha 4\beta 1$ of individual p24-expressing cells based on their proviral integrity. Frequencies of cells expressing high levels of both $\alpha 4$ and $\beta 1$ in each integrity group are indicated at the top.

Reviewer #1 (Remarks to the Author):

the revised manuscript includes changes that answer all of my questions and is much improved.

Reviewer #2 (Remarks to the Author):

Overall, the authors responded to my concerns.

Minor point:

In supplementary figure 5C, one would have liked to see the phenotyping of VLA4 negative CD4 T cell in addition to total and VLA4+ CD4 T cell.

in response to my previous comment "One of these proviruses has lost the entire 5' region to the middle of the pol gene." The authors propose "The most likely explanation for this intriguing finding is an issue at the sequencing step." Given the doubt, the authors should exclude this sequence from the analyses.

Response to the Reviewer's comments

Reviewer #1:

the revised manuscript includes changes that answer all of my questions and is much improved.
We thank the Reviewer for their comment.

Reviewer #2:

Overall, the authors responded to my concerns.

Minor point:

In supplementary figure 5C, one would have liked to see the phenotyping of VLA4 negative CD4 T cell in addition to total and VLA4+ CD4 T cell.

Phenotyping of VLA4 negative cells was added to Supplementary Figure 5c.

in response to my previous comment "One of these proviruses has lost the entire 5' region to the middle of the pol gene." The authors propose "The most likely explanation for this intriguing finding is an issue at the sequencing step." Given the doubt, the authors should exclude this sequence from the analyses.

To address the Reviewer's concern, we excluded this sequence from all analyses. Removing this sequence from the analysis required to adjust all p-values, which were only slightly modified, and did not change any of the original findings.